# Treatment of tubular damage in high-fat-diet-fed obese mice using sodium-glucose co-transporter inhibitors

Sei Saitoh[1]*, Takashi Takaki[2,3], Kazuki Nakajima[4,5], Bao Wo[6,7], Hiroshi Terashima[8], Satoshi Shimo[9], Huy Bang Nguyen[10,11,12], Truc Quynh Thai[10,11,13], Kanako Kumamoto[14], Kazuo Kunisawa[15], Shizuko Nagao[14], Akihiro Tojo[16], Nobuhiko Ohno[17,18], Kazuo Takahashi[1]

1 Department of Biomedical Molecular Sciences (Anatomy II), Fujita Health University School of Medicine, Toyoake, Japan, 2 Department of Anatomy, Showa University School of Medicine, Tokyo, Japan, 3 Center for Electron microscopy, Showa University School of Medicine, Tokyo, Japan, 4 Center for Joint Research Facilities Support, Research Promotion and Support Headquarters, Fujita Health University, Toyoake, Japan, 5 Institute for Glyco-core Research (iGCORE), Gifu University, Gifu, Japan, 6 Department of Anatomy and Molecular Histology, Interdisciplinary Graduate School of Medicine and Engineering, University of Yamanashi, Chuo, Japan, 7 Department of Histology and Embryology, Medical College of Chifeng University, Chifeng, China, 8 JEOL Ltd., Akishima, Japan, 9 Department of Occupational Therapy, Health Science University, Fujikawaguchiko, Japan, 10 Division of Neurobiology and Bioinformatics, National Institute for Physiological Sciences, Okazaki, Japan, 11 Department of Anatomy and Structural Biology, Graduate School of Medical Science, University of Yamanashi, Chuo, Japan, 12 Department of Anatomy, Faculty of Medicine, University of Medicine and Pharmacy (UMP), Ho Chi Minh, Vietnam, 13 Department of Histology Embryology Genetics, Faculty of Basic Medical Sciences, Pham Ngoc Thach University of Medicine, Ho Chi Minh, Vietnam, 14 Education and Research Facility of Animal Models for Human Diseases, Fujita Health University, Toyoake, Japan, 15 Department of Regulatory Science for Evaluation & Development of Pharmaceuticals & Devices, Fujita Health University Graduate School of Health Sciences, Toyoake, Japan, 16 Division of Nephrology & Hypertension, Dokkyo Medical University, Mibu, Japan, 17 Division of Ultrastructural Research, National Institute of Physiological Sciences, Okazaki, Japan, 18 Department of Anatomy, Division of Histology and Cell Biology, Jichi Medical University, Shimotsuke, Japan

* saitoh@fujita-hu.ac.jp

**Data Availability Statement:** The data are all contained within the paper.

## Abstract

A long-term high-fat diet (HFD) causes obesity and changes in renal lipid metabolism and lysosomal dysfunction in mice, causing renal damage. Sodium-glucose co-transporter inhibitors, including phlorizin, exert nephroprotective effects in patients with chronic kidney disease, but the underlying mechanism remains unclear. A HFD or standard diet was fed to adult C57BL/6J male mice, and phlorizin was administered. Lamellar body components of the proximal tubular epithelial cells (PTECs) were investigated. After phlorizin administration in HFD-fed mice, sphingomyelin and ceramide in urine and tissues were assessed and label-free quantitative proteomics was performed using kidney tissue samples. Mitochondrial elongation by fusion was effective in the PTECs of HFD-fed obese mice under phlorizin administration, and many lamellar bodies were found in the apical portion of the S2 segment of the proximal tubule. Phlorizin functioned as a diuretic, releasing lamellar bodies from the apical membrane of PTECs and clearing the obstruction in nephrons. The main component of the lamellar bodies was sphingomyelin. On the first day of phlorizin administration in HFD-fed obese mice, the diuretic effect was increased, and more sphingomyelin was excreted through urine than in vehicle-treated mice. The expressions of three peroxisomal

**Funding:** This work was supported by JSPS KAKENHI [grant numbers 25870281, 16K08439] and the Cooperative Study Program (to S. Saitoh) of the National Institute for Physiological Sciences. The funders had no role in study design, data collection and analysis, decision to publish, or preparation of the manuscript.

**Competing interests:** The authors have declared that no competing interests exist.

β-oxidation proteins involved in fatty acid metabolism were downregulated after phlorizin administration in the kidneys of HFD-fed mice. Fatty acid elongation protein levels increased with phlorizin administration, indicating an increase in long-chain fatty acids. Lamellar bodies accumulated in the proximal renal tubule of the S2 segment of the HFD-fed mice, indicating that the urinary excretion of lamellar bodies has nephroprotective effects.

## Introduction

Western diets high in saturated fatty acids are believed to be a major contributor to the global obesity problem [1]. Centrally obese people are prone to metabolic syndrome, characterized by hypertension, hyperglycemia, and dyslipidemia, and are at a higher risk of cardiovascular disease-related mortality [2]. Obesity and dyslipidemia are major and independent risk factors for chronic kidney disease and end-stage renal disease [3]. Long-term intake of a high-fat diet (HFD) has been reported to not only cause obesity and metabolic syndrome but also induce changes in renal lipid metabolism and lysosomal dysfunction in mice, resulting in renal damage [4]. Sodium-glucose co-transporter (SGLT) inhibitors reduce blood glucose levels by inhibiting the reabsorption of glucose in renal proximal tubular epithelial cells (PTECs) and are used in type 2 diabetes treatment [5]. These inhibitors exert nephroprotective effects in patients with chronic kidney disease, but the mechanism of action is unclear [6]. Therefore, understanding how SGLT inhibitors affect renal lipid metabolism is essential for treating chronic kidney disease. In this study, we investigated the nephroprotective effect of an SGLT inhibitor, phlorizin, against renal injury in a mouse model of HFD-induced obesity.

The kidney has a high-energy demand and abundant mitochondria. Persistent mitochondrial dysfunction causes podocyte injury, tubular epithelial cell damage, and endothelial dysfunction. According to a previous study, megalin-mediated endocytosis of glomerular filtered lipotoxic substances is mainly responsible for damage and autophagy impairment in HFD-induced renal disease [7, 8]. Renal damage in the HFD-induced obesity model is characterized by the accumulation of lamellar bodies, mainly in the S2 segment of the proximal tubule, because megalin is more highly expressed in the S2 segment of microvilli than in the S1 and S3 segments [7, 9]. Lamellar bodies are excreted through the urine of HFD-fed obese mice [10]. Dysfunction of lysosomes leads to decreased autophagic flux, inflammasome activation, and macrophage infiltration [11]. Furthermore, HFD-fed obese mice could not undergo enhanced autophagy activity when they were subjected to pathological stress. Inhibition of autophagy also enhanced HFD-induced mitochondrial dysfunction and inflammasome activation [12]. SGLT inhibitors ameliorate proximal tubular dysfunction in HFD-induced obesity by improving the impairment of autophagic flux [13].

The mitochondrial structure of PTECs is complex; therefore, elucidating their 3D morphology using conventional electron microscopes is difficult. Sphingomyelin, one of the most abundant sphingolipids, plays an important role in cellular and organellar membranes and has been suggested to play a role in lipid rafts. In HFD-fed obese mice, sphingomyelin accumulates in the liver and other organs, and has increased levels in serum [14, 15]. SGLT inhibitors promote diuresis when glucose absorption by PTECs is inhibited, but little is known about the phospholipids excreted through urine. To bridge this gap, we performed serial block-face scanning electron microscopy (SBF-SEM) [16–18], which facilitates efficient acquisition of a series of ultrastructural images. In addition, to investigate components of the lamellar bodies of PTECs and confirm the presence of sphingomyelin, scanning transmission electron

microscope-energy-dispersive X-ray spectroscopy (STEM-EDX) analysis and Raman spectrometry were performed. Liquid chromatography-mass spectrometry (LC-MS) was used to examine sphingomyelin and ceramide in urine, and label-free quantitative proteomics was performed on kidney tissue samples after the administration of SGLT inhibitors in HFD-fed mice.

## Materials & methods

### Animals

All animal experiments were performed in accordance with the National Institutes of Health Guide for the Care and Use of Laboratory Animals (NIH Publications No. 8023, revised 1978). Animal experiments were approved by the Animal Ethics Committees of the University of Yamanashi and Fujita Health University. C57BL/6J mice (No. 000664; Jackson Laboratory, Bar Harbor, ME, USA) were kept under a 12 h light/dark cycle at 24°C and fed a standard low-fat experimental diet *ad libitum*. Four-week-old male mice were randomly divided into two groups. Mice of each group were housed in one cage and fed either the standard diet (STD) (n = 55) or HFD (n = 55) containing 60 kcal of fat, 20 kcal of carbohydrate, and 20 kcal of protein (D12492i; Research Diets, New Brunswick, NJ, USA). They were weighed every four weeks and fed the prescribed diet for 16 weeks. Mice were injected subcutaneously with 400 mg/kg phlorizin (Sigma-Aldrich, St. Louis, MO, USA), which was dissolved in 10% ethyl alcohol, 15% dimethyl sulfoxide, and 75% normal saline (0.9% w/v NaCl) or buffer, and then subjected to two different protocols. Non-fasting blood glucose levels were measured before and after the test injection via tail puncture using a blood glucose meter (Abbott Japan, Tokyo, Japan).

### Phlorizin administration protocol

Protocol 1: Phlorizin or vehicle (n = 34 in each STD and HFD group) was administered to mice at 2 and 16 h prior to perfusion fixation, organ resection, and blood collection. The dosage used in this study was selected based on the results of previous studies [19, 20].

Protocol 2: To reduce the effect of individual differences, the following urine samples were collected consecutively from the same mice in metabolic cages: urine from untreated mice, urine from vehicle-treated mice, urine from day 1 phlorizin-treated mice, and urine from day 2 phlorizin-treated mice. Mice (n = 10 in each STD and HFD group) were housed in metabolic cages for 8 days and had free access to food and water. Drug administration started on day 3 for the vehicle group mice and day 6 for phlorizin or vehicle group mice. The following four types of urine samples were collected: urine from untreated mice (day 3) before vehicle administration, urine from vehicle-treated mice (day 6) before phlorizin or vehicle administration, urine from day 1 phlorizin or vehicle-treated mice (day 7), and urine from day 2 phlorizin or vehicle-treated mice (day 8).

Animals were deeply anesthetized by the intraperitoneal administration of pentobarbital sodium (100 mg/kg body weight, Nacalai Tesque, Kyoto, Japan) for perfusion fixation, blood collection, and organ resection according to protocol 1 or with 2% isoflurane inhalation for organ resection according to protocol 2. The mouse kidneys were kept in Allprotect Tissue Reagent (Qiagen, Hilden, Germany) and stored in a freezer at –30°C.

### Biochemistry

Serum triglyceride and total cholesterol levels were measured (FUJI DRI-CHEM; Fujifilm, Tokyo, Japan). Urinary glucose levels were measured using a blood glucose meter (Abbott,

Japan). Urinary creatinine levels were measured using a LabAssay Creatinine Kit (Fujifilm Wako Pure Chemical Corporation, Osaka, Japan). Urinary protein levels were measured using a Pierce BCA Protein Assay Kit (Thermo Fisher Scientific, Waltham, MA, USA). The insulin in the serum and the liver-type fatty acid-binding protein (L-FABP) in the urine and serum were measured in duplicate using a commercially available enzyme-linked immunoassay kit (#10-1247-01; Mercodia AB, Uppsala, Sweden, and RFBP10; R&D system, Minneapolis, MN, USA).

## Hematoxylin & eosin (H&E) staining and immunostaining of Ki-67

All paraffin-embedded tissues were cut at a thickness of 2–5 μm on a microtome (Yamato Kohki, Saitama, Japan) and mounted on Matsunami Adhesive Slide-coated glass slides (Matsunami Glass, Osaka, Japan). Some sections of immersion or perfusion fixation samples with 4% paraformaldehyde (PFA) were H&E stained, and histomorphology was confirmed by light microscopy. For immunohistochemical staining of Ki-67, tissue sections of perfusion fixation samples (n = 3 in each STD-vehicle, STD-phlorizin, HFD-vehicle, and HFD-phlorizin group) were dewaxed in xylene, ethanol, and phosphate-buffered saline (PBS), followed by autoclave-induced epitope retrieval in 10 mM citrate buffer at 120˚C for 10 min. The samples were permeabilized with 0.1% Triton X 100 in PBS and then incubated with 1% hydrogen peroxide in PBS and 2% gelatin (Sigma-Aldrich, St Louis, MO, USA) in PBS at 25˚C for 1 h each. Immunostaining was performed overnight at 4˚C using an anti-Ki-67 rat monoclonal antibody (SolA15: eBioscience, Thermo Fisher Scientific) in PBS. The same procedure was performed without the primary antibody as a negative control. The specimens were then incubated with HRP-conjugated donkey anti-rat F (ab') 2 antibody (A24543: Invitrogen; Thermo Fisher Scientific) at 25˚C for 1 h and finally visualized with cobalt-enhanced diaminobenzidine in a buffer solution containing hydrogen peroxide (Pierce; Thermo Fisher Scientific) for 5 min. Immunostained sections were additionally incubated with 0.04% osmium tetroxide in 0.1 M phosphate buffer (PB) for 30 s to enhance the contrast of diaminobenzidine-reaction products. Micrographs of Ki-67 nuclei-labeled tubules in the renal cortex were randomly prepared under 20× magnification using an optical BX-61 microscope (Olympus, Tokyo, Japan). The nuclei of Ki-67-positive tubular epithelial cells were counted in the micrographs.

## Fluorescent immunostaining of lysosomal-associated membrane protein 1 (LAMP1) and aquaporin 1

Immunostaining for LAMP1 was used as a lysosomal marker in the proximal tubules. After deparaffinization of the paraffin sections, they were heated at 100˚C for 20 min in 10 mM citrate buffer solution (pH 6.0) for antigen retrieval. The sections were incubated with 4% Block ACE (DS Pharma Biomedical, Osaka, Japan) in PBS (–) containing 0.05% Tween 20 (PBS-T) for 2 h at 25˚C, then washed three times with PBS-T. Primary antibodies against LAMP-1 (ab25245, dilution 1:50; Abcam, Cambridge, UK) and aquaporin 1 (AB2219, dilution 1:300; Merck Millipore) were used as proximal tubule markers. The highly cross-adsorbed secondary antibodies Alexa fluor 488 conjugated donkey anti-rat IgG (H+L) (AB_2535794, 1:500 dilution; Thermo Fisher Scientific) or Alexa fluor 555 conjugated donkey anti-rabbit IgG (H+L) (AB_2535792, 1:500 dilution; Thermo Fisher Scientific) were then used. ProLong™ Diamond Antifade Mountant with 4',6-diamidino-2-phenylindole (DAPI) (Thermo Fisher Scientific) was used for mounting. Fluorescent images were obtained using confocal microscopy (LSM710; Carl Zeiss) equipped with a microscope (Axiovert 200M; Carl Zeiss) with a Plan Apochromat 63×/1.4 NA oil immersion lens, and the images were visualized using the ZEN software (Carl Zeiss).

## Tissue preparation for transmission electron microscopy (TEM), SBF-SEM, and STEM-EDX

Mice (n = 3 in each STD-vehicle, STD-phlorizin, HFD-vehicle, and HFD-phlorizin group) were deeply anesthetized with pentobarbital sodium perfused transcardially with PBS (-) for 30 s and 4% PFA and 1% glutaraldehyde dissolved in 0.1 M PB. After mice were euthanized, perfusion-fixed kidneys were excised, cut into pieces of less than 1 mm using a razor, and incubated overnight at 4˚C with 4% PFA and 1% glutaraldehyde dissolved in 0.1 M PB (pH 7.4). For TEM, tissues were post-treated with 1% osmium tetroxide in 0.1 M PB for 1 h, dehydrated with ethanol in stages, and then embedded in Quetol 812 epoxy resin (Nissin EM, Tokyo, Japan). For SBF-SEM and STEM-EDX, the tissue was fixed using the rOTO method with 2% $OsO_4$ containing 1.5% $K_4[Fe(CN)_6]$ for 1 h on ice, 1% thiocarbohydrazide for 20 min, and 2% $OsO_4$ for 30 min at 25˚C. Subsequently, the tissue was treated with 1% uranyl acetate overnight at 4 ˚C. Tissues were then incubated in a lead aspartate solution for 30 min at 60˚C. After each treatment, tissues were washed four times with double-distilled water. Tissues were dehydrated using a graded series of ethanol (40%, 60%, 80%, 90%, 95%, and 100% for 5 min each) and infiltrated with dehydrated acetone, a 1:1 mixture of resin and acetone, and 100% resin. Specimens were then embedded in Quetol 812 epoxy resin (Nissin EM).

## TEM and data analysis

To examine specimens, sections were first cut at a thickness of 1.0 μm and routinely stained with toluidine blue. Micrographs of proximal convoluted tubules and glomeruli in the outer cortex were randomly prepared under 60× magnification using an optical BX-61 microscope (Olympus). Injured proximal renal tubules were counted in the micrographs. Ultrathin sections were cut at a thickness of 70–80 nm with a diamond knife on an ultramicrotome, mounted on copper grids, and double-stained with uranyl acetate and lead citrate. The sections were finally observed using a transmission electron microscope at an accelerating voltage of 80 kV (JEM-1400 flash; JEOL, Tokyo, Japan).

## SBF-SEM and data analyses

Epon blocks were trimmed and mounted on aluminum rivets using a conductive adhesive (Chemtronics, Kennesaw, GA, USA). Surfaces of trimmed specimens were sputtered with gold to increase conductivity, and specimens were observed using a MERLIN or SIGMA/VP SEM system (Carl Zeiss AG, Jena, Germany) equipped with a 3 View in-chamber ultramicrotome (Gatan Inc., Pleasanton, CA, USA). The images of observed samples were captured under various imaging conditions by microscopy. Serial section images from SBF-SEM of mouse proximal tubules were taken at every 50-nm thickness over 500 slices and processed using Fiji/ImageJ (http://fiji.sc/wiki/index.php/Fiji) incorporated with TrakEM2 [21] (Cardona A, et.al. 2012), Amira (Thermo Fisher Scientific), and Simpleware (Synopsys, Mountain View, CA, USA), and then reconstructed into 3D images. Mitochondria were randomly selected in each segment of the proximal tubule. The mitochondrial area was measured using the area list and the polyline tool in the TrakEM2 software. Mitochondrial volume was calculated using TrakEM2.

## STEM-high-angle annular dark-field (HAADF) images and EDX

EDX maps and HAADF images were acquired using an FEI Tecnai Osiris transmission electron microscope equipped with a Super X EDX detector (Thermo Fisher Scientific) to

investigate elemental distribution. Ultrathin sections of 70–80 nm thickness of the same block after SBF-SEM data acquisition were used for observation.

## Raman spectrometry and data acquisition

Raman spectra were measured using a JEOL JRS-SYSTEM 1000 Raman spectrometer (JEOL) in the range of 200–4,000 cm$^{-1}$ (Raman shift) excited by a 532 nm YAG laser at 20 mW output power using a 50× objective lens (Fig 6E and 6F). The irradiated laser spot diameter at the sample point was approximately 2 μm. Perfusion fixation of kidney samples was performed with 4% PFA in PBS (-), and the samples were embedded in an optimal cutting temperature (OCT) compound (Sakura Finetek USA, Inc., Torrance, CA, USA). Frozen tissue sections were cut into pieces of 10 μm thickness using a microtome cryostat (MICROM International GmbH, Walldorf, Germany). The OCT compound was removed by soaking in PBS for at least 30 min. Sphingomyelin ($C_{39}H_{79}N_2O_6P$) (Funakoshi Co., Ltd., Tokyo, Japan) and PFA were used for comparison.

## Sphingolipid and ceramide measurement

Total lipids were extracted from urine (200 μL) and kidney tissues (20–40 mg) supplemented with 10 pmol Sphingolipid Mixture II (Avanti Polar Lipids, Alabaster, AL, USA), as mentioned previously [22]. Total lipid extracts were further hydrolyzed in 2 mL of 0.1 M potassium hydroxide (Fujifilm Wako) in chloroform/methanol (2:1, v/v) for 3 h at room temperature, neutralized with acetic acid (Fujifilm Wako), and partitioned according to the Folch method [23]. The lower phase was collected and dried under a stream of nitrogen. Dried lipids were dissolved in 1 mL of mobile phase A (acetonitrile/water/formic acid, 97:2:1, v/v/v, with 5 mM ammonium formate), and a 2-μL aliquot was injected into the LC-MS system using a triple quadrupole mass spectrometer LCMS8060 coupled to a Nexera X2 liquid chromatography system (Shimadzu Corp., Kyoto, Japan). A hydrophilic interaction chromatography-MS/MS system, as previously described, was used [24]. Each peak was integrated using the Lab Solution Insite software and quantified based on the peak areas of the standards. The sphingomyelin amount was normalized against the internal standard (SM d18:1/17:0) recovery data and expressed as pmol/creatine.

## Label-free quantitative proteomics

For proteomic analysis, kidney samples from STD-vehicle (n = 4), HFD-vehicle (n = 4), and HFD-PLZ (n = 4) mice were extracted using phase-transfer surfactant (10 mM sodium deoxycholate, 10 mM sodium lauroyl sarcosinate, 0.1 M Tris–HCl, pH 9.0) buffer. at 95˚C for 5 min. The samples were sonicated using a BioRaptor (BM Equipment Co., Ltd., Tokyo, Japan) with 40 cycles of 30 s sonication and 15 s off-time. The protein concentration of the samples was measured using a micro-BCA kit (Thermo Fisher Scientific). Reduction and alkylation of the samples were performed with 10 mM dithiothreitol (Thermo Fisher Scientific) at 25˚C for 30 min and with 20 mM iodoacetamide (Thermo Fisher Scientific) at 25˚C for 30 min in the dark. Then, samples were quenched with 5 mM dithiothreitol (Thermo Fisher Scientific) at 25˚C for 10 min. Each sample was processed for proteomic analysis according to the solid phase enhanced sample preparation (SP3) protocol [25]. Sera-Mag SpeedBeads (GE Healthcare, Madison, WI, USA) were added to the sample. Then, the sample coupled with Sera-Mag SpeedBeads was washed three times with 80% ethanol. Next, ammonium bicarbonate (0.1 M, pH 8) containing 0.2 μg/L Trypsin/rLys-C Mix (Promega, Madison, WI, USA) was added and gently extracted into each tube at a ratio of 1:50 (enzyme to protein). The tubes were then incubated at 37˚C for 14 h using a thermomixer with shaking. After digestion, the sample was

acidified with trifluoroacetic acid (Kanto Kagaku Co., Ltd., Japan) (v/v) and desalted with GL-Tip-SDB (GL Science). The peptides in the samples were analyzed using direct LC-MS using an Orbitrap Fusion Tribrid mass spectrometer (Thermo Fisher Scientific) coupled to an EASY-nLC 1000 liquid chromatography system (Thermo Fisher Scientific).

### Bioinformatics and statistical analysis of proteomics data

To screen for differentially expressed proteins (DEPs) in the maximum range and minimize the loss of meaningful targets, DEPs were defined as those with fold change (FC) >1.2 or <1/1.2 ($P < 0.05$ based on a $t$-test.) Gene Ontology (GO) annotation and Kyoto Encyclopedia of Genes and Genomes (KEGG) pathway analysis were performed using Protein Discoverer 2.4 (Thermo Fisher Scientific). KEGG pathways were used to analyze fatty acid and sphingolipid metabolism.

### RNA extraction and quantitative real-time reverse transcription PCR (qRT-PCR)

Total RNA was extracted using Trizol RNA Separation Reagent and the RNeasy Mini Kit according to the manufacturer's protocol (Qiagen, Valencia, CA, USA). First-strand cDNA was synthesized using ReverTra Ace qPCR RT Master Mix (Toyobo, Osaka, Japan). For quantitative PCR, THUNDERBIRD SYBR qPCR Mix (Toyobo) was used, and real-time PCR quantification was performed using a StepOne Real-Time PCR System (Life Technologies, Carlsbad, CA, USA). Quantitative PCR analysis was performed using a StepOne analyzer (Life Technologies). The qPCR data were measured four times per sample. The data were obtained by the calibration method, and the mean value was calculated. The left side of each figure shows the relative expression levels of each gene in comparison to those of *Gapdh*. The housekeeping gene *Gapdh* was used to normalize all PCR data. All PCR primer sequences are described in S1 Table in S1 File.

### Statistics

Statistical analyses were performed on GraphPad Prism version 6 for Windows (GraphPad Software Inc., La Jolla, CA, USA) using unpaired two-tailed Student's $t$-test, paired two-tailed Student's $t$-test, one-way analysis of variance (ANOVA) followed by the post-hoc Tukey–Kramer tests, Kruskal–Wallis post-hoc tests for multiple comparisons (Dunn test), two-way ANOVA followed by the post-hoc Tukey–Kramer test, and Sidak's multiple comparison test. Proteome analysis was performed with Proteome Discoverer 2.4 (Thermo Fisher Scientific) using a two-tailed Student's $t$-test. Significant differences were determined as $P < 0.05$. All values are expressed as the mean (standard deviation) or median (interquartile range) for normally and non-normally distributed data, respectively.

## Results

### Obesity, prediabetes, and phlorizin-responsive hyperglycemia in 60% HFD-fed mice

After 16 weeks of feeding (Fig 1A), HFD-fed mice showed characteristic symptoms of prediabetes, including weight gain, abnormal blood glucose levels, and hyper insulin levels in the serum under non-fasting conditions, compared with STD-fed mice. To observe the progression of obesity induced by the HFD feeding, the body weights of the mice in the HFD- and STD-fed groups were measured every four weeks. During the study period, the weight gain of the HFD-fed mice was greater than that of the STD-fed mice, and the weights of the two

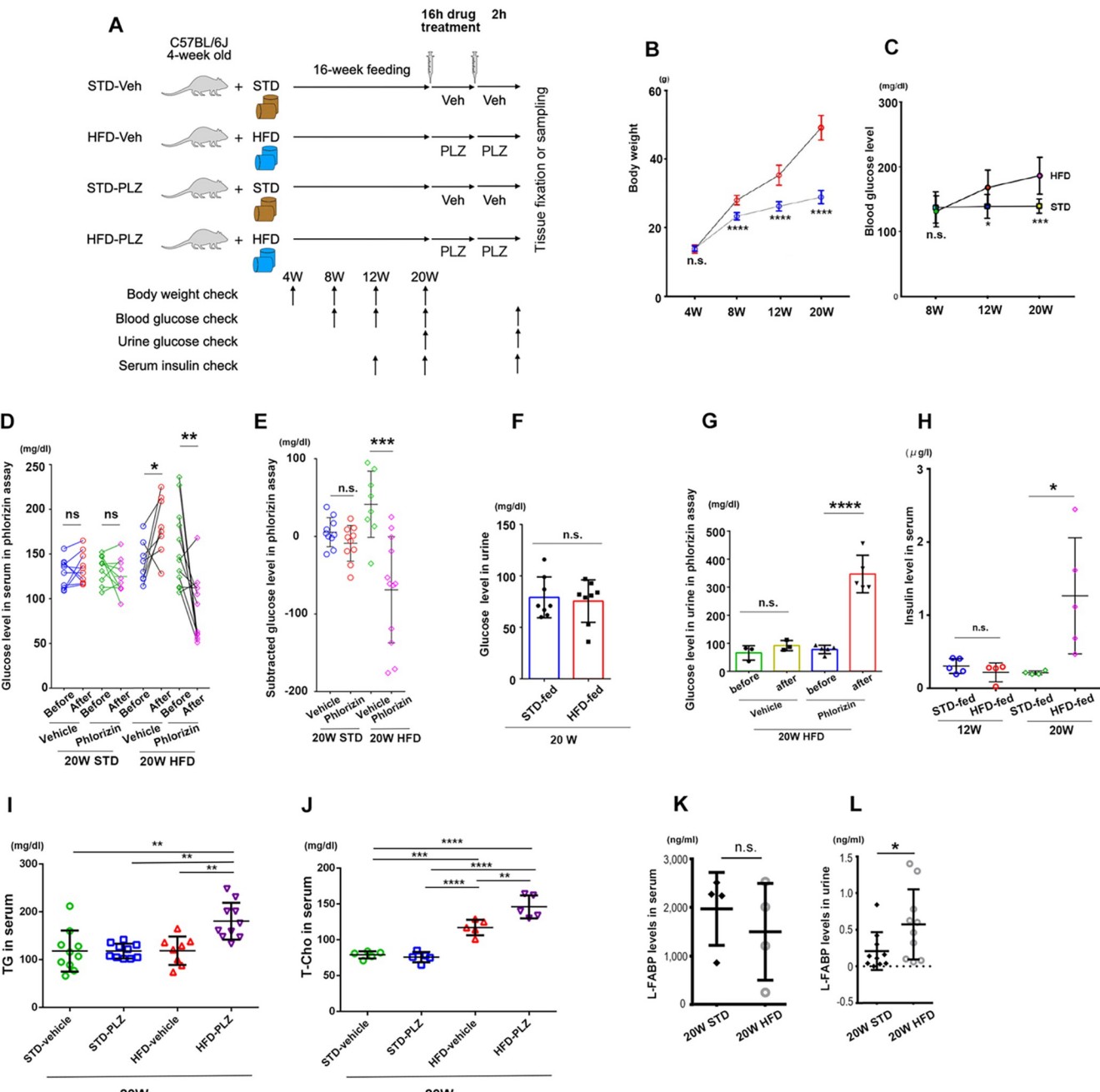

**Fig 1. Phlorizin improved increased blood glucose levels in high-fat diet (HFD)-fed mice.** (A) Experimental design: All data were measured under non-fasting conditions. (B) Weight gain in HFD-fed and standard diet (STD)-fed mice (n = 10) over 16 weeks. (C) Blood glucose levels (n = 10). (D) Blood glucose levels pre- and post-vehicle (Veh) administration as well as pre- and post-phlorizin (PLZ) administration in STD- and HFD-fed mice at 20 weeks of age. (E) Subtracted glucose levels from the data shown in (D) before and after vehicle or phlorizin administration. (F) Urinary glucose levels in STD- and HFD-fed mice at 20 weeks of age. (G) Urinary glucose levels pre- and post-Veh as well as pre- and post-PLZ administration in HFD mice at 20 weeks of age. (H) Serum insulin levels in STD- and HFD-fed mice at 12 and 20 weeks of age. (I) Serum triglyceride (TG) and (J) total cholesterol (T-CHO) in the serum of STD- and HFD-fed mice at 20 weeks of age. (K) Serum L-FABP levels in STD- and HFD-fed mice at 20 weeks of age. (L) Urinary L-FABP levels in STD- and HFD-fed mice at 20 weeks of age. Results are presented as the mean ± standard deviation (SD). $^*P < 0.05$, $^{**}P < 0.01$, $^{***}P < 0.001$, $^{****}P < 0.0001$. Statistical significance was calculated by unpaired two-tailed Student t-test (B, C, E, F, G, H, K, and L), paired two-tailed Student t-test (D), and one-way analysis of variance, followed by the post-hoc Turkey–Kramer tests (I, J).

groups were significantly different after four weeks ($P < 0.0001$). Differences persisted up to 20 weeks of age (Fig 1B; 8–20 weeks of age). Blood glucose levels were higher in HFD-fed mice than in the STD-fed mice at 12 and 20 weeks of age (12 weeks: $P < 0.05$, 20 weeks: $P < 0.001$) (Fig 1C). Phlorizin administration at 20 weeks of age significantly suppressed glucose levels in HFD-fed mice but not in STD-fed mice ($P < 0.001$) (Fig 1D and 1E). Urine glucose levels were not significantly different between STD- and HFD-fed mice at 20 weeks of age (Fig 1F). After phlorizin administration, urine glucose levels were significantly increased in HFD-fed mice at 20 weeks of age ($P < 0.0001$), but they were unchanged in HFD-fed mice with vehicle administration (Fig 1G). Serum insulin levels were not significantly different at 12 weeks of age and were significantly higher albeit showing heterogeneity in HFD-fed mice than in STD-fed mice at 20 weeks under non-fasting conditions ($P < 0.05$) (Fig 1H). The results indicate that the onset of obesity occurred after 8–20 weeks of HFD intake and was accompanied by an increase in blood glucose levels, which was ameliorated by phlorizin administration. The HFD-fed mice at 20 weeks of age were in an early diabetic state and had elevated insulin secretion. The heterogeneity of insulin secretion by the HFD-fed mice at 20 weeks of age may be influenced by the *ad libitum* consumption of the HFD diet at night and differences in mouse-specific insulin resistance [26]. It is also possible that elevated postprandial blood glucose levels affect insulin secretion. Phlorizin administration significantly increased serum triglyceride and total cholesterol levels in HFD-fed obese mice ($P < 0.01$) (Fig 1I and 1J). In addition, urinary L-FABP levels, a marker of proximal tubular injury, were significantly higher in the HFD-fed mice at 20 weeks of age than in the STD-fed mice, whereas there were no changes in the serum levels ($P < 0.05$; Fig 1K and 1L).

## Injured proximal convoluted tubules accumulated lamellar bodies in HFD-fed mice, which was ameliorated by the administration of phlorizin

Kidney tissues were observed by light and electron microscopy (Fig 2A–2L), and a remarkable difference in the proximal tubules of HFD-non-drug-treatment and HFD-vehicle mice was observed (Fig 2A–2D, 2G and 2K). Because damage to the proximal convoluted tubule was easily observed in the toluidine blue section (Fig 2G and 2H, red arrows), lamellar bodies were identified in the damaged area of the proximal tubule by electron microscopy (Fig 2K and 2L, indicated by red arrows). The number of damaged tubules per high power field in toluidine blue (TB) sections of four groups (STD-vehicle, STD-phlorizin, HFD-vehicle, and HFD-phlorizin) was compared (Fig 2E–2H). In the mice from STD-vehicle and -phlorizin groups, tubular damage and lamellar bodies were absent (Fig 2E, 2F, 2I and 2J). The incidence of proximal convoluted tubular injury was significantly higher in HFD-vehicle mice than in HFD-phlorizin mice ($P < 0.05$), suggesting that phlorizin administration improved the proximal convoluted tubular injury (Fig 2Q). In the proximal tubules of STD-vehicle mice (S1A–S1D Fig in S1 File), similar to STD-PLZ mice, immunolocalization of LAMP1 (lysosomal marker) was observed in the apical dots (S1E, S1F Fig in S1 File). However, in HFD-vehicle mice (S1I–S1L Fig in S1 File), the immunoreactivity of LAMP1 in the apical region was markedly reduced and stained in lamellar bodies (Herzig MC, et al. 2011). Conversely, in HFD-PLZ mice (S1M–S1Q Fig in S1 File), the immunolocalization of LAMP1 in the proximal tubules was sparsely scattered, with numerous vacuoles. To investigate the effect of damaged proximal tubules on cell division, immunostaining was performed using the Ki-67 antibody, a marker of cell proliferation (Fig 2M–2P). The number of Ki-67 in HFD-vehicles was significantly higher than that in the STD-vehicle, STD-PLZ, and HFD-PLZ mice ($P < 0.0001$) (Fig 2R). Phlorizin administration in HFD-induced obese mice decreased the proliferative capacity of proximal tubules.

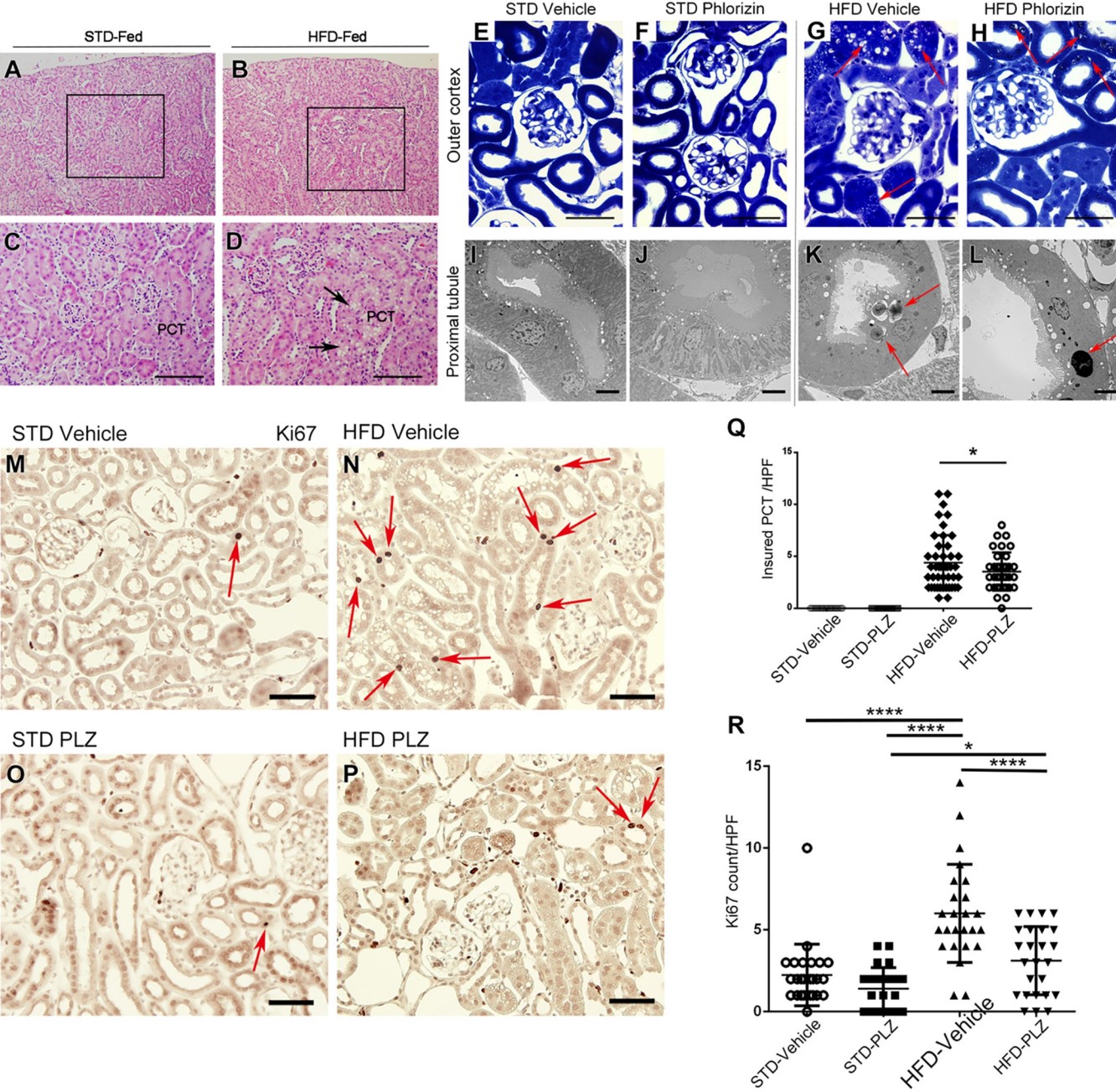

**Fig 2. Phlorizin improved injured proximal tubules in high-fat diet (HFD)-fed mice.** (A–D) Light micrographs of 2-μm paraffin kidney sections stained with hematoxylin and eosin (H&E) in standard diet (STD)-fed and HFD-fed mice. (E–H) Light micrographs of 1-μm thick Epon kidney sections stained with toluidine blue (TB). The TB sections have glomeruli and renal tubules at the outer cortex of four groups: (E) STD-vehicle, (F) STD-phlorizin (PLZ), (G) HFD-vehicle, and (H) HFD-PLZ (G–H). Red arrows indicate damaged proximal tubules in HFD-vehicle and HFD-PLZ mice. (I–L) Electron micrographs of proximal tubules of four groups; (I) STD-vehicle, (J) STD-PLZ, (K) HFD-vehicle, and (L) HFD-PLZ. (K and L) The high intensity of the lamellar body is indicated by red arrows corresponding to the red arrows in (G) and (H). (M–P) Light micrographs of immunohistochemistry for Ki-67 in the renal cortex of four groups; (M) STD-vehicle, (N) STD-PLZ, (O) HFD-vehicle, and (P) HFD-PLZ. Ki-67-positive tubular nuclei are indicated by red arrows. (Q) The number of injured proximal tubules per high-magnified light micrograph of TB sections (E–H). (R) The number of Ki-67-positive nuclei in the proximal tubules per high-power light micrograph of immunohistochemistry. Results are presented as the mean ± standard deviation, $^*P < 0.05$, $^{****}P < 0.0001$. Statistical significances were calculated by unpaired two-tailed Student's t-test (Q) and one-way analysis of variance, followed by the post-hoc Turkey–Kramer tests (R). Bars; (C and D) 200 μm, (E–H) 50 μm, (I–L) 5 μm, (M–P) 50 μm.

## Accumulated phosphorus in lamellar bodies revealed by elemental mapping using STEM-EDX

STEM HADDF electron micrographs showed lamellar bodies with interwoven lipid membranes in the S2 segment of the proximal tubules of HFD-vehicle mice (Fig 3A and 3B). In the STEM HADDF mode, lipid membranes and mitochondrial cristae were visible. We performed elemental mapping analysis using EDX to analyze components accumulated in the lamellar bodies, which showed an accumulation of phosphorus, osmium, and lead (Fig 3C, 3G and 3H). Osmium and lead were accumulated in the lamellar bodies in rOTO fixation and en bloc staining (Fig 3G–3I). The phosphorus accumulated in the lamellar bodies in the form of phospholipids, which are an integral component of their membranes (Fig 3C–3F).

## SBF-SEM revealed the mitochondrial structure of proximal tubules

Proximal tubules were morphologically divided into three segments (S1, S2, and S3) in STD-vehicle mice [27], and their morphological features were redefined using SBF-SEM (Fig 4A and 4B).

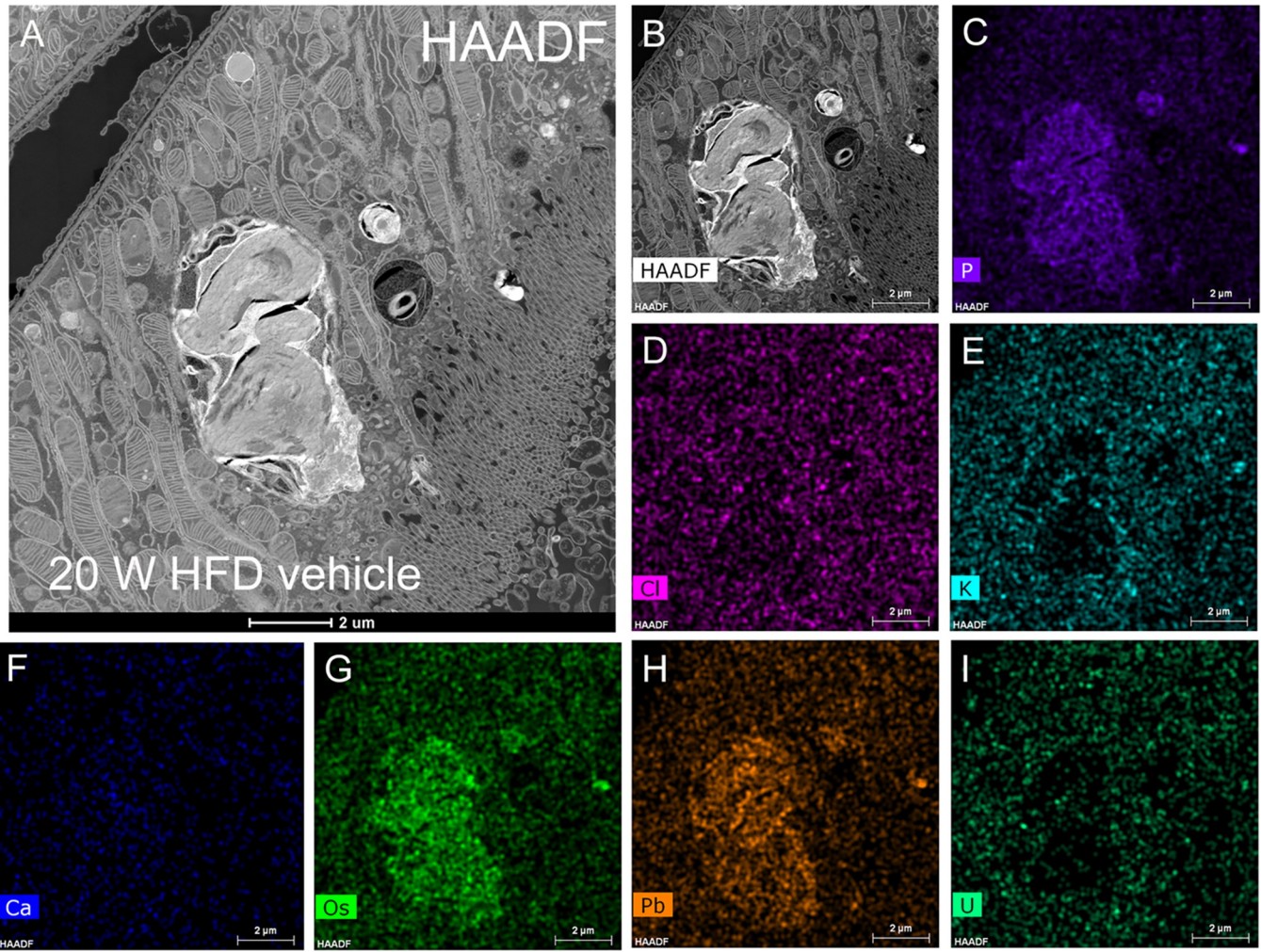

**Fig 3. Scanning transmission electron microscope-energy-dispersive X-ray spectroscopy (STEM)-EDX analysis of lamellar bodies for elementary mapping.** (A–B) STEM-high-angle annular dark-field (HAADF) images show lamellar bodies in the proximal tubule of high-fat diet (HFD)-fed vehicle mice at 20 weeks. (C–I) EDX elementary mapping images. Each image has the atomic number in the upper left corner and the element name in the lower-left corner. (C) Phosphorus; atomic number 15. (D) Chlorine; atomic number 17. (E) Potassium; atomic number 19. (F) Calcium; atomic number 20. (G) Osmium; atomic number 76. (H) Lead; atomic number 82. (I) Ulan; atomic number 92.

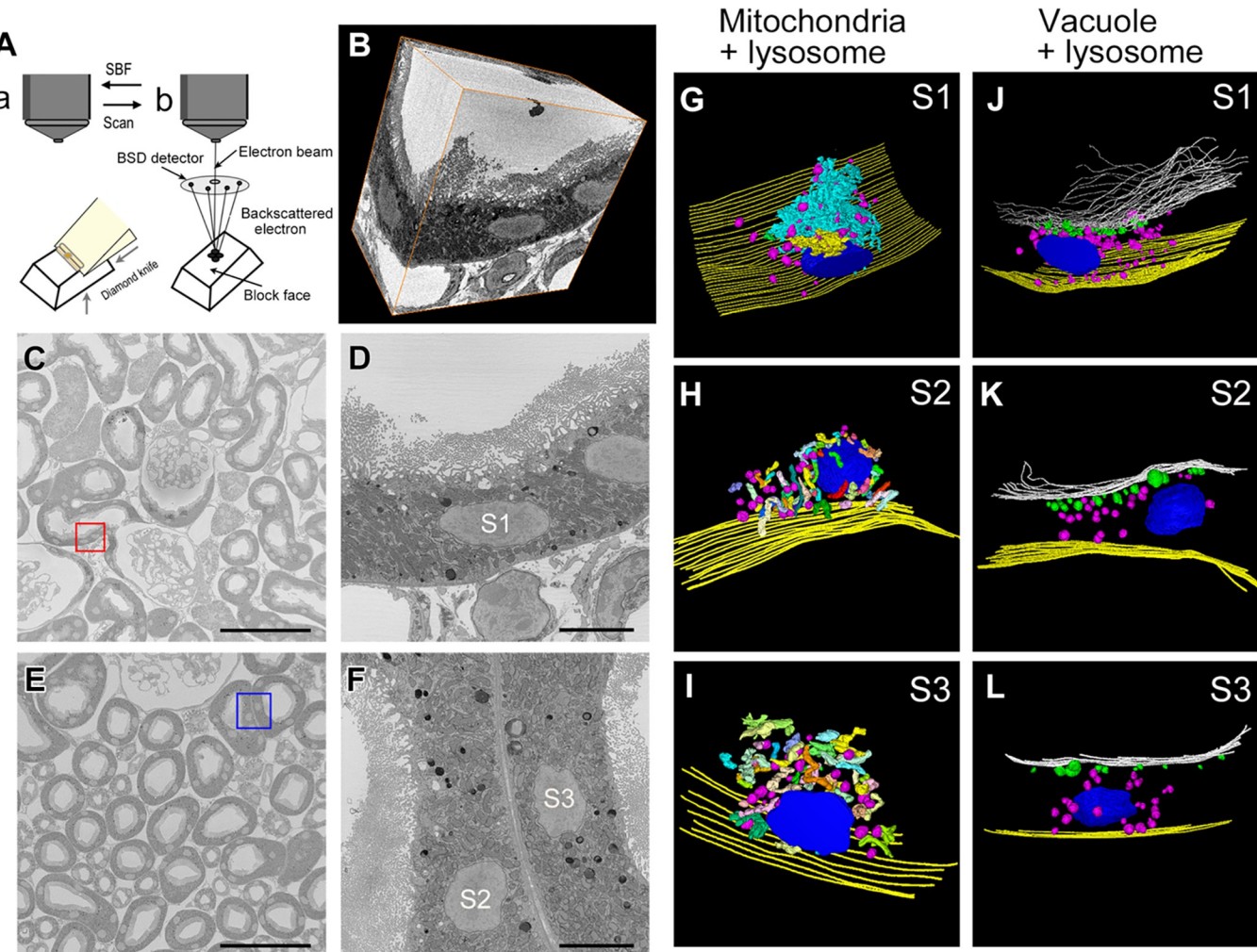

**Fig 4. Electron microscopic 3D reconstruction showing the proximal tubule as a characteristic mitochondrial shape in three segments (S1, S2, and S3 segments).** (A) For scanning electron microscopy (SEM), a diamond knife (yellow) was used to cut ultrathin slices from the top of the block-faces of the Epon block (a). Serial block-face SEM (SBF-SEM) can produce a series of hundreds of serial electron micrographs of the block surfaces similar to conventional transmission electron microscopy (TEM) images (b). After the ultrathin sections are cut, the SBF-SEM data are acquired for backscattered electrons using a backscatter detector. (B) Three-dimensional reconstruction of serial SBF-SEM images of proximal tubule segment 1. (C) Lower magnification SBF-SEM image of the outer cortex in the kidney of an STD-vehicle mouse. (D) Higher magnification image of proximal tubule segment 1 in the red box in (C). (E) Lower magnification SBF-SEM image of the outer medulla in the kidney of an STD-vehicle mouse. (F) Higher magnification image of proximal tubule segments 2 and 3 in the blue box in (E). (G–I) Reconstructed 3D image of mitochondria (various colors other than magenta) and lysosomes (magenta). The yellow line along the basement membrane is shown in steps of 500 nm (0.5 um) for every 10 slices. (G) Image of SBF-SEM data corresponding to the S1 segment shown in (D). (H and I) Image of SBF-SEM data corresponding to the S2 and S3 segments shown in (F). (J–L) Reconstructed 3D image of vacuoles (green) and lysosomes (magenta). The yellow line along the basement membrane is shown in steps of 500 nm (0.5 μm) for every 10 slices. The white line along the apical surface of tubular cells (brush border-bottom line) is shown in steps of 500 nm (0.5 μm) for every 10 slices. (J) Image of SBF-SEM data corresponding to the S1 segment shown in (D). (K and L) Image of SBF-SEM data corresponding to the S2 and S3 segments shown in (F). Bars; (C and E) 100 μm, (D and F) 50 μm.

PTECs of the S1 segment showed a tall brush border, a well-developed endocytic lysosome, numerous elongated mitochondria, and extended basolateral invaginations and interdigitations (Fig 4C and 4D). Notably, the 3D reconstruction data of the S1 segment mitochondria were large and complex, showing a meshwork structure of connecting rod and branch mitochondria (Fig 4G). The PTEC in the S2 segment was similar to that in the S1 segment, but the brush border was shorter, and the basolateral invagination and interdigitation were less prominent. The PTEC in the S3 segment was more cuboidal and with fewer endocytic organelles and inconspicuous

membrane invagination and interdigitations (Fig 4E and 4F). In the 3D reconstruction, the mitochondria of the S2 and S3 segments were small and simple structures, such as rod-shaped and branched, compared with those of the S1 segment (Fig 4H and 4I). Vacuoles were abundant in the apical region, whereas lysosomes were abundant from the apical to the basal regions (Fig 4J–4L).

## Fragmented mitochondria in proximal tubules were increased in HFD-fed mice and were elongated by phlorizin administration

To examine cellular damage in the proximal tubules at the organelle level, SBF-SEM analysis was performed in three segments (S1, S2, and S3) in the HFD obese mouse model following phlorizin administration (Fig 5A–5L). Giant mitochondria ($>4 \times 10^{10}$ nm³) were observed in

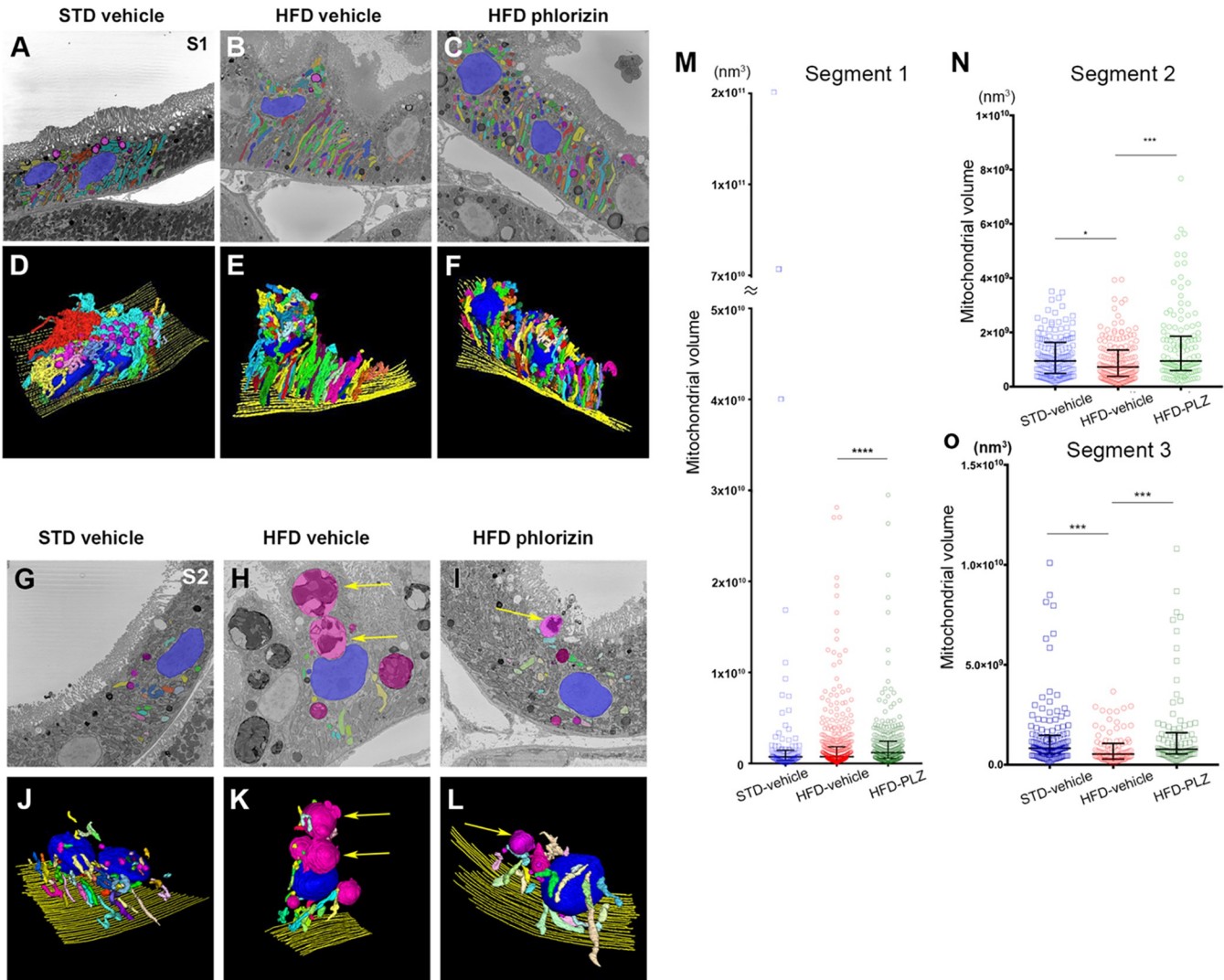

**Fig 5. Electron microscopic 3D reconstruction of the HFD phlorizin assay model revealed mitochondrial volume changes.** (A–C) Segmentation of mitochondria (various colors other than magenta) and lysosomes (magenta) on serial block-face (SBF)-SEM 2D images of the proximal tubule S1 segment. (D–F) Three-dimensional reconstruction of serial SBF-SEM images of the proximal tubule S1 segment. (A, D) Standard diet (STD) vehicle, (B, E) high-fat diet (HFD) vehicle, and (C, F) HFD phlorizin. (G–I) Segmentation of mitochondria (various colors other than magenta), lysosomes (magenta), and lamellar bodies (magenta; yellow arrows) on SBF-SEM 2D images of the proximal tubule S2 segment. (J–L) Three-dimensional reconstruction of serial SBF-SEM images of proximal tubule S2 segment. (G, J) STD-vehicle, (H, K) HFD-vehicle, and (I, L) HFD phlorizin. (M–O) Statistical analyses of mitochondrial volume in SBF-SEM reconstruction data. Results are presented as median with an interquartile range (M–O). *$P < 0.05$, ***$P < 0.001$, ****$P < 0.0001$. Statistical significance was calculated by the Kruskal–Wallis test with the post-hoc test, followed by Dunn's multiple tests. PLZ, phlorizin.

the STD-vehicle mice but not in the S1 segment of the HFD-vehicle and HFD-phlorizin mice (Fig 5D–5F). Fragmented mitochondria appeared to be gathered at the top of the nucleus, pushing up the brush border at the apical surface in the S1 segment of the HFD-vehicle mice (Fig 5B and 5E). In the S2 segment of the HFD-vehicle mice, the mitochondria were fragmented, and lamellar bodies were present at the top of the nucleus pushing up the brush border (Fig 5H and 5K). The volume of mitochondria in the S2 segment of the HFD-vehicle mice was lower than that in the STD-vehicle mice (Fig 5J, 5K and 5N). In HFD-phlorizin mice, the number of lamellar bodies decreased in the proximal tubules of the S2 segment (Fig 5K and 5L). The volume of mitochondria in the mice fed HFD-phlorizin was higher than in the fragmented mitochondria of S1, S2, and S3 segments of HFD-vehicle mice (Fig 6M–6O). In the HFD-phlorizin mice, in some proximal tubules of the S2 segment, vacuolar components were formed both apically and basally. The basal vacuole was large and longitudinal in shape, resembling the ER-Golgi morphology of the proximal tubule under ER stress (S2 Fig in S1 File) [28].

## Identification of sphingomyelin in lamellar bodies of proximal convoluted tubules of HFD-vehicle mice using Raman spectrometry

Raman spectrometry was used to measure the glomeruli and spherical particles (lamellar bodies) of the proximal convoluted tubule in the outer layers of the cortex, and the S3 segment of the proximal tubule in the outer layers of the medulla, to obtain different Raman spectra (Fig 6A–6G). In the Raman spectra of spherical particles (lamellar bodies) in the PCT, the spectra of all the components were detected in situ (Fig 6G and 6H). Subtraction of the S3 segment of the Raman spectrum of the proximal tubule as an internal control from the Raman spectrum of the spherical body in PCT was necessary to observe lamellar bodies only (Fig 6H–6J). Raman spectra were measured at five different points on each of the spherical particles (lamellar body) and the S3 segment of the proximal tubule (Fig 6H and 6I). The spectra of the five locations at the S3 site had approximately the same profile (Fig 6I). The third spectrum had the highest S/N in the S3 segment of the proximal tubule (Fig 6I3). The calculation was performed by subtracting the third spectrum of the S3 segment of the proximal tubule from that of the five spherical particles. The shift (Fig 6J H2–I3) of the subtracted Raman peak was similar to that of the Raman spectrum of the purified sphingomyelin. Raman spectrometry confirmed the presence of sphingomyelin in the lamellar body.

## Sphingomyelin excretion was increased in HFD obese mice on the first day of phlorizin treatment, but ceramide was not affected

Urine test results showed a diuretic effect and an increase in urinary protein levels on the first day of phlorizin treatment in both STD and HFD mice and vice versa on the second day of phlorizin treatment (Fig 7B–7E).

The amount of sphingomyelin and ceramide in the urine excreted by HFD-fed obese mice after phlorizin administration was quantified by liquid chromatography-tandem mass spectrometry. Interestingly, in HFD-fed mice, the total amount of sphingomyelin consisting of sphingosine (d18:1) and sphinganine (d18:0) was significantly higher on the first day of phlorizin administration than that in the vehicle administration group ($P < 0.05$) (Fig 8A, S3 Fig in S1 File). When STD- and HFD-fed mice were compared, the total amount of urinary sphingomyelin was significantly higher in HFD-fed mice with phlorizin administration on day 1 and day 2 than in STD-fed mice with phlorizin administration on day 1 and day 2, respectively (d18:0: day 1; $P < 0.05$, day 2; $P < 0.01$), (d18:1: day 1; $P < 0.01$, day 2; $P < 0.05$) (Fig 8A). In contrast, ceramide, the precursor of sphingomyelin biosynthesis, was excreted through urine at a lower level than sphingomyelin. The total excretion of ceramide was not significantly

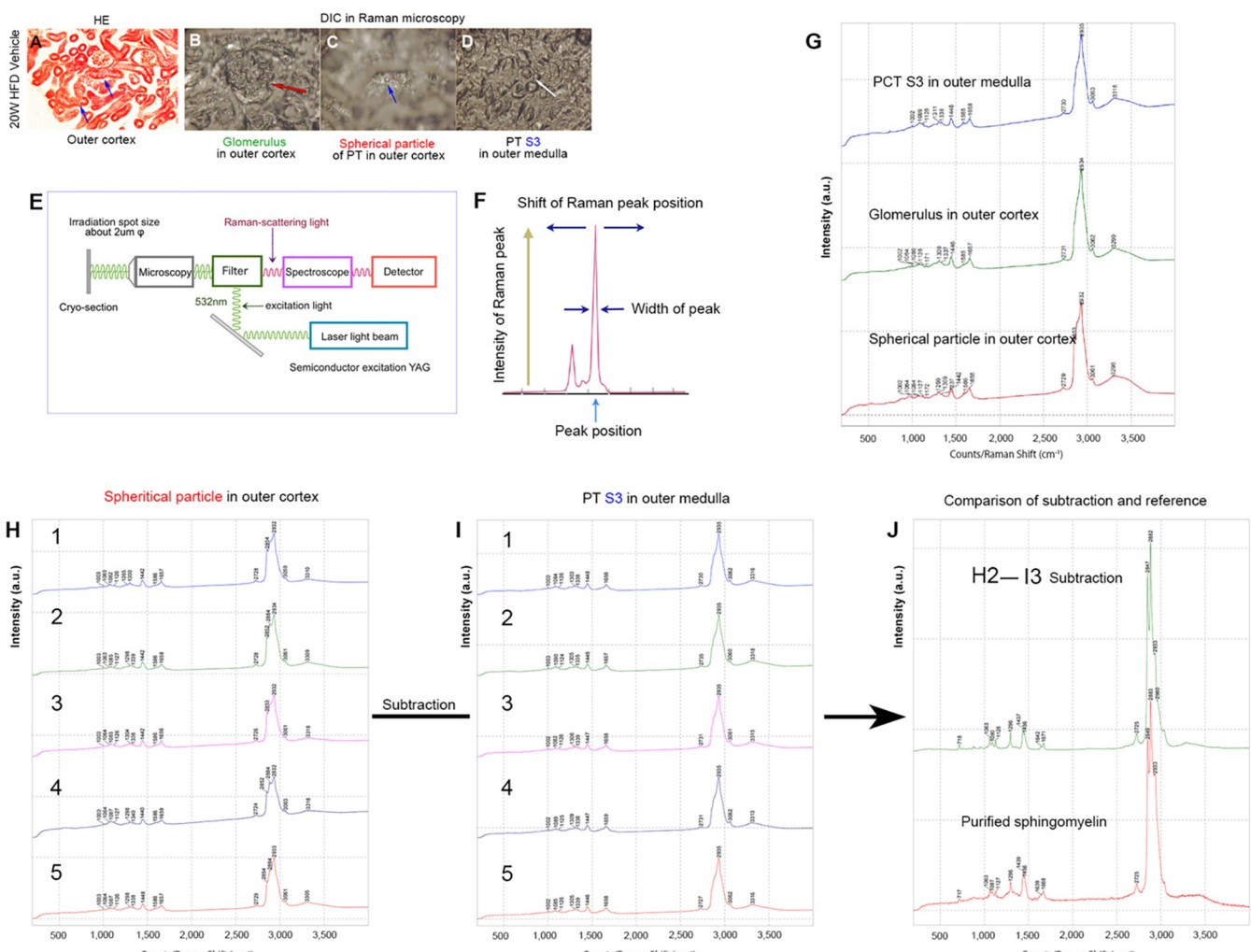

**Fig 6. Raman spectrometry showed that lamellar bodies (spherical particles) in the proximal tubule in HFD-vehicle mice are sphingomyelin.** (A) Hematoxylin and eosin (H&E) staining of the frozen sections. Spherical particles (blue arrow) correspond to lamellar bodies in EM images. (B–D) Differential interference contrast (DIC) images of Raman microscopy. (B) Glomerulus (red arrow) in the outer cortex, (C) spherical particle (blue arrow) of the proximal tubule (PT) in the outer cortex, (D) PT segment 3 (S3) (white arrow) in the outer medulla. (E) Schematic diagram of the detection system of a Raman microscope. (F) Name of each part of the Raman spectrum. (G) Raman spectra of PT S3 in the outer medulla (blue), glomerulus (green), and the spherical particle of PT in the outer cortex (red). (H) Five-point variation of Raman spectra of spherical particles in PT in the outer cortex. (I) Five-point variation of Raman spectra of PT S3 in the outer medulla. (J) The spectra of the spherical particles and segment S3 were compared, and the spectrum at 2800–3000 cm$^{-1}$ derived from the methyl/methylene group differed (H, I). A prominent peak was observed at 2800–3000 cm$^{-1}$, suggesting that the subtracted Raman peak has more chain hydrocarbon structures. In addition, differential peaks were observed below 1700 cm$^{-1}$, and weak P = O bonds were observed at 1063 cm$^{-1}$ and 1296 cm$^{-1}$, suggesting that spherical particles (lamellar body) might contain phosphate. Two main types of hydrocarbons were seen: saturated hydrocarbons (such as palmitic acid) and unsaturated hydrocarbons (such as oleic acid). The shift (H2–I3) of the subtracted Raman peak was similar to that of the Raman spectrum of the purified sphingomyelin.

altered by phlorizin administration (Fig 8B, S3 Fig in S1 File). Furthermore, the total amount of sphingomyelin was not different among the four groups (STD-vehicle, STD-phlorizin, HFD-vehicle, and HFD-phlorizin) (Fig 8C). These data indicate that the amount of sphingomyelin excreted through urine did not affect the total amount of sphingomyelin in the kidney tissue. In contrast, in HFD-vehicle mice, the total amount of ceramide in tissue consisting of sphingosine (d18:1) and sphinganine (d18:0) was significantly higher than that in STD-vehicle mice (d18:0: $P < 0.01$, d18:1: $P < 0.05$) but not significantly different from that in mice administered HFD-phlorizin (Fig 8D).

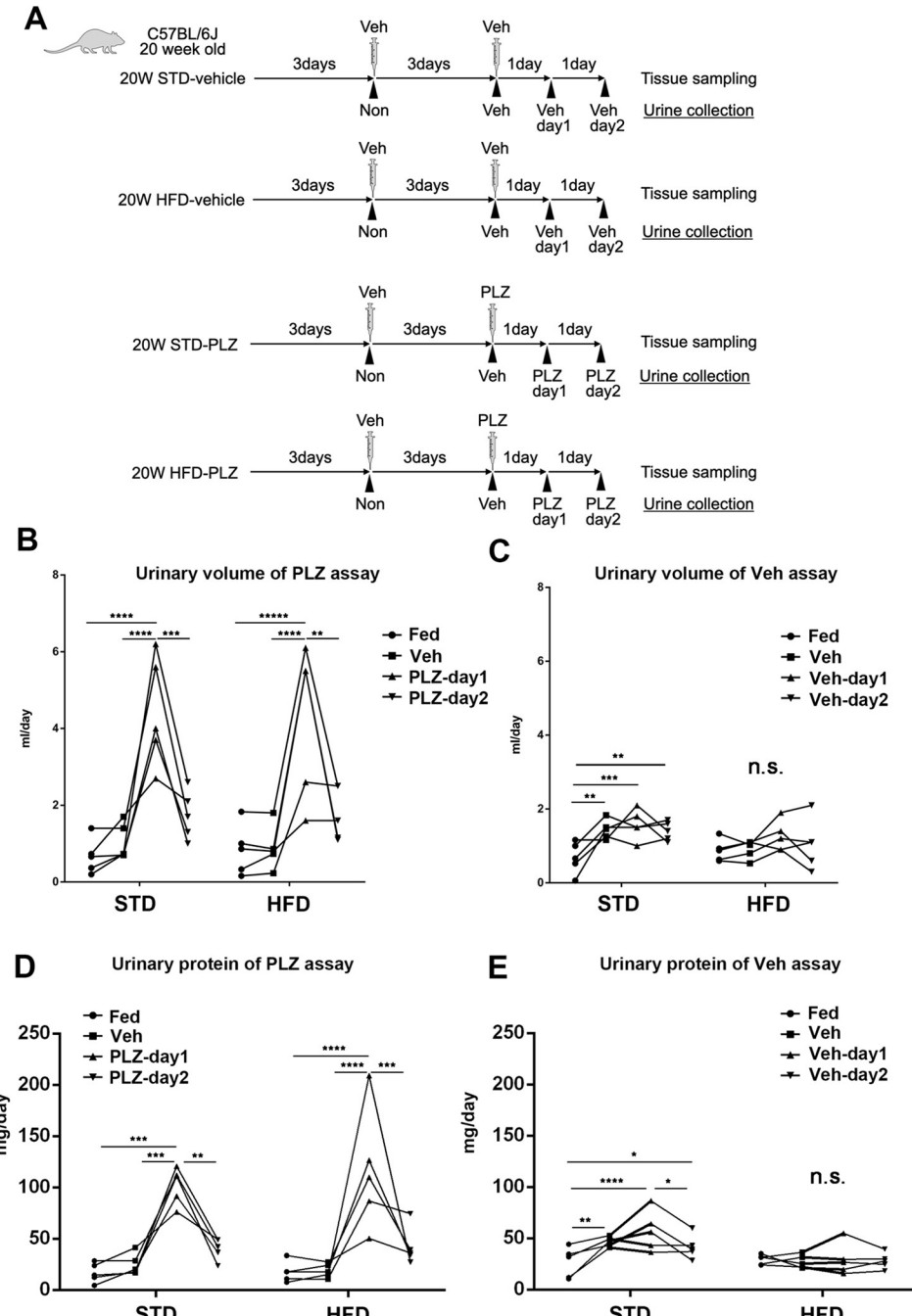

**Fig 7. Protocols for urine and tissue collection for sphingomyelin and ceramide analysis and changes in urine volume and proteinuria.** (A) Diagram showing urine collection and tissue sampling scheduled after administration of phlorizin (PLZ) or vehicle (Veh) to C57/BL6J mice fed with a standard diet (STD) or high-fat diet (HFD) at 20 weeks. Urine was collected in the absence of dosing (Fed); 3 days after administration of Veh; after the second administration of Veh (Veh day 1, Veh day 2); and after administration of PLZ (PLZ day 1, PLZ day 2). (B) The urinary volume of PLZ assay. (C) The urinary volume of Veh assay. (D) The urinary protein of PLZ assay. (E) The urinary protein of Veh assay. $^{*}P < 0.05$, $^{**}P < 0.01$, $^{***}P < 0.001$, $^{****}P < 0.0001$. Statistical significances were calculated by two-way analysis of variance followed by the post-hoc Turkey–Kramer tests.

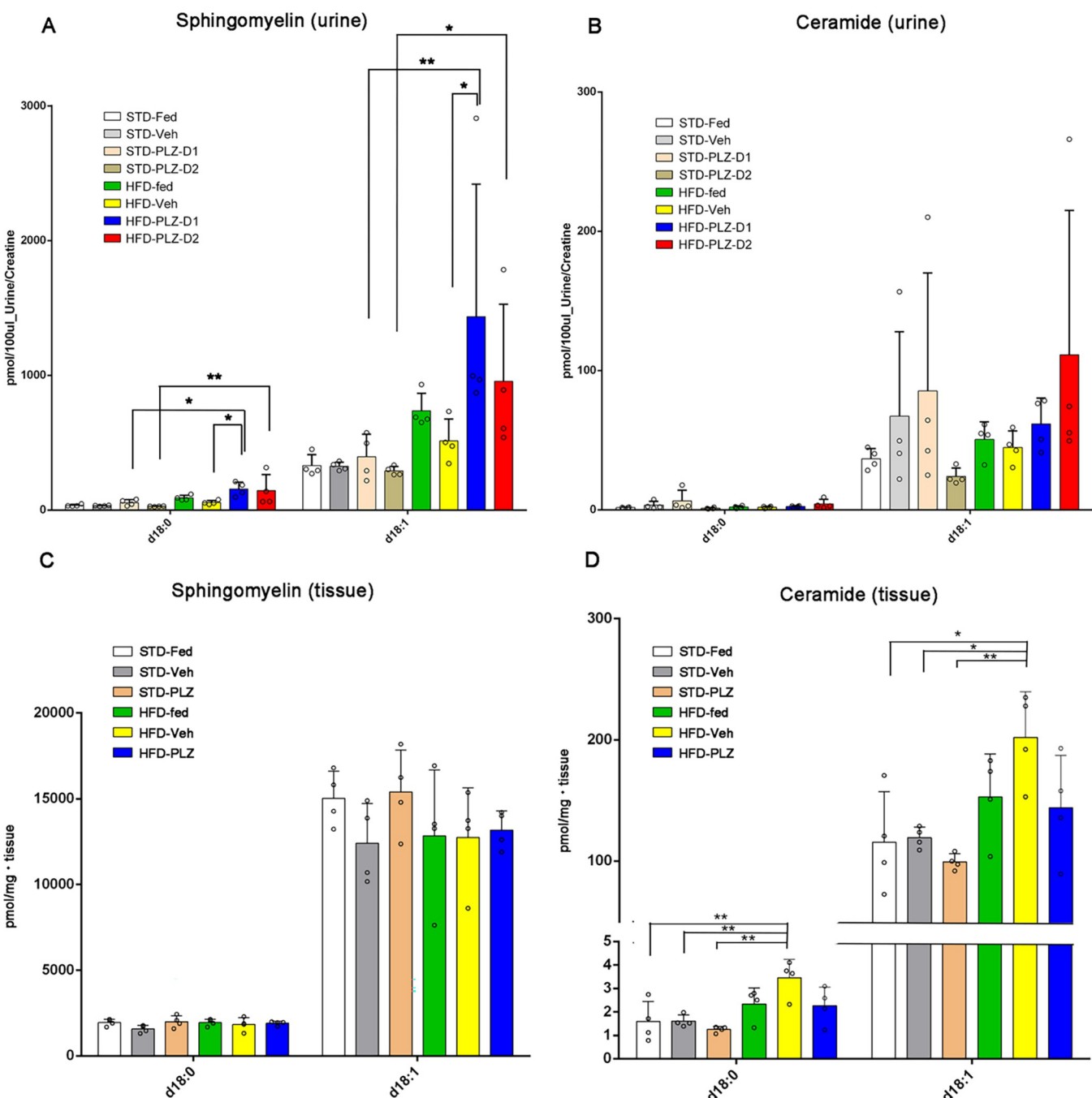

**Fig 8. Sphingomyelin levels in urinary extracts and those in tissues from the kidneys of standard diet (STD)- and high-fat diet (HFD)-fed mice in the phlorizin assay.** Liquid chromatography-tandem mass spectrometry was used to quantify sphingomyelin levels in urinary extracts from (A) total sphingomyelin (d18:0), sphingomyelin (d18:1), (B) total ceramide (d18:0), and ceramide (d18:1) in the HFD obese mouse model in the phlorizin assay. (C) Total sphingomyelin (d18:0), sphingomyelin (d18:1), (D) total ceramide (d18:0), and ceramide (d18:1) of tissues in the HFD obese mouse model from the phlorizin assay. Results are presented as the mean ± standard deviation. $^*P < 0.05$, $^{**}P < 0.01$. Statistical significances were calculated by two-way analysis of variance followed by the post-hoc Turkey–Kramer tests and Sidak's multiple comparison test. Urine was obtained from C57/BL6J mice fed with a STD (STD-Fed) or HFD (HFD-Fed) prior to vehicle administration, vehicle-treated mice (Veh) prior to phlorizin or vehicle administration, phlorizin- or vehicle-treated mice on day one (PLZ-D1 or Veh-D1), and phlorizin- or vehicle-treated mice on day two (PLZ-D2 or Veh-D2).

## Peptide and protein identification

In total, 4,177 proteins were identified, with 179 DEPs (72 increased and 107 decreased regulatory proteins) significantly altered between HFD-vehicle and HFD-PLZ according to thresholds of FC > 1.2 or < 1/1.2 and $P < 0.05$. According to the GO annotation, among the biological processes, the expression of proteins was related to transport, stress response, other metabolic processes, other biological processes, cell signaling, and protein metabolism processes (S6A and S6D Fig in S1 File). In terms of molecular function, protein expression was related to kinase, enzyme regulator, transporter, and nucleic acid-binding activities (S6B and S6E Fig in S1 File). Regarding cellular components, protein expression was related to other cellular components, plasma membrane, other membrane, mitochondria, and ER-Golgi (S6C and S6F Fig in S1 File). KEGG pathway annotation was used to screen for proteins involved in fatty acid metabolism, the TCA cycle, and sphingolipid metabolism. Fatty acid synthesis (*Fasn*) was downregulated, and the TCA cycle (*Suclg2*) was upregulated in STD-vehicle vs. HFD-vehicle mice (S2 Table in S1 File). Fatty acid denaturation was classified as upregulated significantly in ω oxidation (*Cypa4a10* and *Cypa4a14*) and downregulated in peroxisomal β oxidation (*Acox1*, *Acox3*, *Ecl3*, *Acaa1b*, and *Ehhadh*) (Table 1, S6G Fig in S1 File). Fatty acid elongation was upregulated significantly in *Acot2* and *Ppt1*. Sphingolipid metabolism was absent in DEPs for all 13 species detected, including *Smpd2*, *Cers2*, *Cers6*, *Asah1*, and *Asah2*.

**Table 1. Fatty acid degradation.**

| Gene name | Abundance Ratio | | Abundance Ratio Adj. P-Value | |
|---|---|---|---|---|
| | HFD-PLZ/HFD-Veh | HFD-Veh/TD-Veh | HFD-PLZ /HFD-Veh | HFD-Veh/STD-Veh |
| Eci1 | 1.168 | 1.035 | N.S. | N.S. |
| Cyp4a10 | 1.449 | 1.333 | 0.00083325 | 0.0425003 |
| Cyp4a14 | 1.9 | 1.158 | 3.1872E-09 | N.S. |
| Cyp4a12a | 1.095 | 1.387 | N.S. | 0.03310635 |
| Acox1 | 0.762 | 1.321 | 0.00112547 | N.S. |
| Acox3 | 0.762 | 1.363 | 0.00108359 | 0.04065159 |
| Cpt1a | 0.986 | 1.131 | N.S. | N.S. |
| Cpt1b | 1.306 | 0.773 | N.S. | N.S. |
| Cpt2 | 1.024 | 1.107 | N.S. | N.S. |
| Gcdh | 1.04 | 1.021 | N.S. | N.S. |
| Acadvl | 1.174 | 1.001 | N.S. | N.S. |
| Acadl | 1.052 | 1.051 | N.S. | N.S. |
| Eci2 | 0.872 | 1.019 | N.S. | N.S. |
| Eci3 | 0.804 | 0.99 | 0.01816364 | N.S. |
| Acaa1a | 0.888 | 1.074 | N.S. | N.S. |
| Acaa1b | 0.821 | 0.958 | 0.03506118 | N.S. |
| Acads | 0.952 | 1.069 | N.S. | N.S. |
| Ehhadh | 0.762 | 0.875 | 0.00112547 | N.S. |
| Acat1 | 1.041 | 0.9 | N.S. | N.S. |
| Acat2 | 1.039 | 1.07 | N.S. | N.S. |
| Acadsb | 1.109 | 0.761 | N.S. | N.S. |
| Acadm | 0.958 | 0.946 | N.S. | N.S. |

Statistical analyses of abundance ratio were performed using two-tailed Student's *t*-test. Significant differences were determined as $P < 0.05$. N.S, Not significant in Abundance Ratio Adj. P-Value; PLZ, phlorizin; Veh, vehicle.

### Sphingomyelinase (Smpd1/2) and sphingomyelin synthesis (Sgms1/2) mRNA levels in an HFD mouse phlorizin administration model

The *Smpd1* mRNA levels of HFD-fed mice were significantly lower than those of STD-fed mice ($P < 0.05$) (S4A Fig in S1 File). In contrast, the mRNAs levels other than *Smpd2*, *Sgms1*, and *Sgms2* were not significantly different between STD-fed mice and HFD-fed mice (S4 Fig in S1 File). The phlorizin assay showed no significant changes in the gene expression levels of sphingomyelinase (*Smpd1/2*) and sphingomyelin synthesis (*Sgms1/2*) among the four groups (STD-vehicle, STD-phlorizin, HFD-vehicle, and HFD-phlorizin) (S4 Fig in S1 File).

## Discussion

### Phlorizin administration elongated the fragmented mitochondria of PTECs in the S1–S3 segments of the proximal tubules in HFD-fed obese mice

In SBF-SEM, the 3D reconstructed data of mitochondria of PTECs in the S1 segment of STD-vehicle mice showed large meshwork structures of more than $4 \times 10^{10}$ nm$^3$. The mitochondrial structure of PTECs in the S2 and S3 segments of the proximal tubule in the STD-vehicle mice was simple, small, branched, and rod-shaped. Phlorizin administration elongated the fragmented mitochondria of PTECs in the S1–S3 segments of the proximal tubules in HFD-fed obese mice [13, 29]. SGLT2 was localized in the S1/S2 segment and SGLT1 in the S3 segment of the proximal tubule. Phlorizin administration increased mitochondrial elongation by SGLT1 and SGLT2 inhibitors in the PTECs of S1–S3 segments in HFD obese mice [30, 31]. Mitochondrial elongation by fusion was effective in the PTECs of HFD-fed obese mice following phlorizin administration. Impairment of the mitophagy clearance mechanism of injured mitochondria resulted in the accumulation of fragmented mitochondria in the renal cortex of diabetic kidneys [32]. Excessive fragmentation of mitochondria during fission results in dysfunction of autolysosome formation and accumulation of lamellar bodies in HFD-fed obese mice [4, 12]. A decrease in mitochondrial function promotes reactive oxygen species generation, activation of apoptotic pathways, increased inflammatory responses, and endoplasmic reticulum stress [33]. Urinary metabolomic signatures can detect mitochondrial dysfunction in diabetic kidney disease [34]. In HFD-fed obese mice, phlorizin administration significantly increased serum triglyceride and total cholesterol levels after a short administration. On the other hand, clearance of triglycerides and total cholesterol is known to be increased by SGLT inhibitors [35, 36]. These results are consistent with the fact that in C57BL/6J HFD-fed mice, phlorizin supplementation decreases plasma cholesterol and triglycerides after long-term administration [37]. SGLT inhibitors normalize glucose metabolism and suppress oxidative stress in diabetic mouse kidneys [38].

### Phlorizin administration induced the release of sphingomyelin identified in lamellar bodies in HFD-fed obese mice

Renal intravital images of HFD-fed obese mice have shown accumulation of lamellar bodies in the PTECs of the S2 segment of the proximal tubule and local narrowing of the lumen of the tubules and peritubular capillaries [7]. Capillaries around the lumen between vacuolated PTECs were also narrowed and blood cell flow was disrupted in HFD-fed obese mice, which was rarely observed in STD-fed mice [7]. In this study, many lamellar bodies were found in the apical portion of the S2 segment of the proximal tubules of HFD-fed obese mice. Secretory granule marker (LAMP1)-positive lamellar bodies are excreted through the urine of HFD-fed obese mice [10, 39]. Phlorizin administration may function as a diuretic by releasing lamellar

bodies from the apical membrane of the PTECs and clearing the obstruction of the nephron in HFD-fed obese mice. SGLTs caused an excessive influx of sodium ions into PTECs with glucose when blood glucose levels were high. Excessive sodium ions were actively excreted out of the PTECs by $Na^+–K^+$ ATPase. We hypothesized that SGLT inhibitor-induced exocytosis of lamellar bodies normalizes the membrane potential abnormalities in the PTECs of HFD-fed obese mice [40]. SGLT2 is predominantly expressed in the S2 segment, where lamellar bodies are abundant in HFD-fed mice, and it is possible that phlorizin mainly affected SGLT2 to induce the release of lamellar bodies from the apical membrane of PTECs. In HFD-phlorizin mice, endoplasmic reticulum stress was induced as the apical and basal vacuoles increased in size and number. The endoplasmic reticulum stress induced by the efflux of cytoplasm membrane components has been shown to cause mitochondrial elongation and compensate for the loss of apical membranes, such as lipid rafts [41]. Raman spectrometry showed that the main component of the lamellar bodies is sphingomyelin [42]. Lamellar bodies are lysosome-related secretory organelles found in type II lung alveolar cells and skin keratinocytes [43]. These lamellar bodies contain phospholipids, glucosylceramides, sphingomyelin, and cholesterol [44]. Long-chain sphingomyelin and ceramide levels were increased in patients with diabetic kidney disease, suggesting that the sphingomyelin-ceramide pathway is involved in renal complications [45]. Imaging mass spectrometry showed an accumulation of sphingomyelin (d18:/16:0) in the glomeruli, arteries, and renal veins of HFD-fed obese mice [46]. To quantitatively examine lamellar body excretion in urine, we measured sphingomyelin and ceramide in the urine of HFD-fed obese mice administered with phlorizin by LC-MS. When sphingomyelin identified in the lamellar body was excreted through urine, fragmented mitochondria were elongated, and the cell proliferation marker Ki-67 was decreased in the proximal tubule.

### Three peroxisomal β-oxidation proteins of fatty acid degeneration were downregulated, and two fatty acid elongation proteins were upregulated in the HFD-PLZ assay

As the amount of sphingomyelin in urine increased, we examined the gene expression levels of sphingomyelinase and sphingomyelin synthesis, which affect the amount of sphingomyelin produced in kidney tissues. *Smpd1* mRNA levels in the entire kidneys of HFD-nondrug-treated mice were lower than those in STD-non-drug-treated mice. However, there was no difference in the levels of *Smpd1*, *Smpd2*, *Sgms1*, and *Sgms2* mRNA or the significant genes responsible for sphingomyelin and ceramide turnover in the entire kidneys of the HFD-fed obese mice with phlorizin administration. Proteomic analysis showed no predominant changes in sphingolipid metabolism owing to phlorizin administration in HFD mice. In contrast, the three peroxisomal β-oxidation proteins of fatty acid degeneration were downregulated [47]. Two fatty acid elongation proteins were upregulated in the HFD-PLZ assay, indicating an increase in long-chain fatty acids. One possible limitation of this study is that since the SGLT inhibitors were not selective, the inhibition of SGLT1 and SGLT2 in the proximal tubules occurred simultaneously. In contrast, making the SGLT inhibitors non-selective allowed us to evaluate systemic obesity and its effects on the entire renal tubules. This will help us design new SGLT inhibitors that can act on fatty acid metabolism in obesity in the future.

## Conclusions

SGLT inhibitors effectively treat renal damage in patients with type 2 diabetes. Numerous lamellar bodies were found in the apical portion of the S2 segment of the proximal tubule of HFD-fed obese mice. Sphingomyelin, the main component of lamellar bodies, was excreted through urine during phlorizin (SGLT inhibitor) administration in HFD-fed mice. Our study

shows that lamellar bodies that accumulated in the S2 segment of HFD-fed mice and the urinary excretion of sphingomyelin identified in lamellar bodies have nephroprotective effects.

## Supporting information

**S1 File. Contains all the supporting tables and figures.**
(DOCX)

## Acknowledgments

We would like to thank Zheng Huang and Jiaorong Chen of the Department of Anatomy and Molecular Histology, Interdisciplinary Graduate School of Medicine and Engineering, University of Yamanashi, Japan, for their technical support.

## Author Contributions

**Conceptualization:** Sei Saitoh, Takashi Takaki, Nobuhiko Ohno.

**Data curation:** Sei Saitoh, Kazuki Nakajima, Hiroshi Terashima, Akihiro Tojo, Nobuhiko Ohno.

**Funding acquisition:** Sei Saitoh.

**Investigation:** Sei Saitoh, Takashi Takaki, Kazuki Nakajima, Bao Wo, Hiroshi Terashima, Satoshi Shimo, Huy Bang Nguyen, Truc Quynh Thai, Kanako Kumamoto, Kazuo Kunisawa, Nobuhiko Ohno.

**Writing – original draft:** Sei Saitoh, Shizuko Nagao, Akihiro Tojo, Nobuhiko Ohno, Kazuo Takahashi.

**Writing – review & editing:** Sei Saitoh, Takashi Takaki, Kazuki Nakajima, Bao Wo, Hiroshi Terashima, Satoshi Shimo, Huy Bang Nguyen, Truc Quynh Thai, Kanako Kumamoto, Kazuo Kunisawa, Shizuko Nagao, Akihiro Tojo, Nobuhiko Ohno, Kazuo Takahashi.

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
