## [Decision Letter · Decision Letter 0]

6 Nov 2022

PONE-D-22-21058Treatment of tubular damage in high-fat-diet-fed obese mice using sodium-glucose co-transporter inhibitorsPLOS ONE

Dear Dr. Saitoh,

Thank you for submitting your manuscript to PLOS ONE. After careful consideration, we feel that it has merit but does not fully meet PLOS ONE’s publication criteria as it currently stands. Therefore, we invite you to submit a revised version of the manuscript that addresses the points raised during the review process.

Please address all comments of reviewerof both reviewers.==============================

We look forward to receiving your revised manuscript.

Kind regards,

Ines Armando, PhD

Academic Editor

PLOS ONE

“Yes.This work was supported by JSPS KAKENHI [grant numbers 25870281, 16K08439] and the Cooperative Study Program (to S. Saitoh) of the National Institute for Physiological Sciences.”

“This work was supported by JSPS KAKENHI [grant numbers 25870281, 16K08439] and the Cooperative Study Program (to S. Saitoh) of the National Institute for Physiological Sciences.”

“Yes.This work was supported by JSPS KAKENHI [grant numbers 25870281, 16K08439] and the Cooperative Study Program (to S. Saitoh) of the National Institute for Physiological Sciences.”

Reviewers' comments:

Reviewer's Responses to Questions

**Comments to the Author**

1. Is the manuscript technically sound, and do the data support the conclusions?

Reviewer #1: Yes

Reviewer #2: Yes

2. Has the statistical analysis been performed appropriately and rigorously? 

Reviewer #1: Yes

Reviewer #2: Yes

3. Have the authors made all data underlying the findings in their manuscript fully available?

Reviewer #1: Yes

Reviewer #2: Yes

4. Is the manuscript presented in an intelligible fashion and written in standard English?

Reviewer #1: Yes

Reviewer #2: Yes

5. Review Comments to the Author

Reviewer #1: With the increase in obesity and obesity-related chronic kidney disease incidence, the role of new therapies that confer nephroprotection are essential. In that sense, Saitoh et al. investigated in detail the underlying mechanisms behind the nephroprotective effect of phlorizin, a sodium-glucose co-transporter (SGLT) inhibitor. The manuscript is well-written and detailed. By using different techniques to elucidate the 3D mitochondrial structure of proximal tubular epithelial cells (PTECs), as well as to investigate components of lamellar body components in those cells, the authors demonstrated that phlorizin releases lamellar bodies from the apical membrane of PTECs, which reduces proximal tubular injury (as assessed by the analysis of L-FABP urinary levels). I believe that some points should be taken in consideration:

Introduction:

- Lines 150 and 151: At the end of the Introduction section, the authors stated, “Understanding how SGLT inhibitors affect renal lipid metabolism is essential for treating chronic kidney disease.”. I believe this phrase could be best used if it was in the first paragraph of the Introduction.

Results:

- Figure 1H: In Figure 1H, the authors use “STD-non” and “HFD-non” abbreviations. The meaning of those abbreviations is not presented in the legend of the Figure. I suppose the correct abbreviations should be “STD-fed” and “HFD-fed”. Moreover, is there an explanation for the large variance noticed in the “HFD-non” group, which is not perceived in the other groups?

- Lines 411 and 412: The authors demonstrated that phlorizin administration significantly increased serum triglyceride and total cholesterol levels in HFD-fed obese mice. Nevertheless, phlorizin supplementation has shown to reduce plasma cholesterol and tryglycerides in C57BL/6J HFD-fed mice in a previously published paper (Liu et al., 2021). I believe the authors should discuss these differences in the discussion.

Reference: Liu D, Ji Y, Guo Y, et al. Dietary Supplementation of Apple Phlorizin Attenuates the Redox State Related to Gut Microbiota Homeostasis in C57BL/6J Mice Fed with a High-Fat Diet. J Agric Food Chem. 2021;69(1):198-211. doi:10.1021/acs.jafc.0c06426

- Figure 8: To avoid confusion, I would suggest the authors to use similar colors for the same groups in Figure 8A and 8C. For instance, while blue is used for a HFD-PLZ group in Figure 8A, it is used for a HFD-vehicle group in Figure 8C. Using similar colors could avoid interpretation mistakes by the readers.

Discussion:

- Lines 827 to 829: The authors refer that one possible limitation of the study was that phlorizin is a non-selective SGLT inhibitor. Nonetheless, which SGLT do the authors believe is more related to the releasing of lamellar bodies from the apical membrane of PTECs?

Reviewer #2: Reviewers Comments

Manuscript ID: PONE-D-22-21058

Study Title: Treatment of tubular damage in high-fat-diet-fed obese mice using sodium-glucose co transporter inhibitors

Authors: Saitoh et al.

The study entitled “Treatment of tubular damage in high-fat-diet-fed obese mice using sodium-glucose co-transporter inhibitors” by Saitoh et al aims to explore underlying nephroprotective effects mechanism of Sodium-glucose co-transporter inhibitors. The study is well designed and robust which shows the nephroprotective effects of lamellar bodies accumulated in the S2 segment and urinary excretion of sphingomyelin. The methodology is well described and results are well presented and discussed.

There are few minor concerns which needs to be taken care.

Minor Comments:

1. Page 11, line 173: Please reframe the sentence for clarity.

2. The manuscript is overall too lengthy and needs to be concised wherever possible.

6. PLOS authors have the option to publish the peer review history of their article (what does this mean?). If published, this will include your full peer review and any attached files.

Reviewer #1: **Yes: **Pedro Alves Soares Vaz de Castro

Reviewer #2: No

---

## [Author Response · Author response to Decision Letter 0]

20 Dec 2022

Review Comments to the Author

Reviewer #1：Comments

With the increase in obesity and obesity-related chronic kidney disease incidence, the role of new therapies that confer nephroprotection are essential. In that sense, Saitoh et al. investigated in detail the underlying mechanisms behind the nephroprotective effect of phlorizin, a sodium-glucose co-transporter (SGLT) inhibitor. The manuscript is well-written and detailed. By using different techniques to elucidate the 3D mitochondrial structure of proximal tubular epithelial cells (PTECs), as well as to investigate components of lamellar body components in those cells, the authors demonstrated that phlorizin releases lamellar bodies from the apical membrane of PTECs, which reduces proximal tubular injury (as assessed by the analysis of L-FABP urinary levels). I believe that some points should be taken in consideration:

Introduction:

- Lines 150 and 151: At the end of the Introduction section, the authors stated, “Understanding how SGLT inhibitors affect renal lipid metabolism is essential for treating chronic kidney disease.”. I believe this phrase could be best used if it was in the first paragraph of the Introduction.

Thank you for your suggestion. This phrase has been moved to P6, Lines 84–86 of the first paragraph of the Introduction.

Results:

- Figure 1H: In Figure 1H, the authors use “STD-non” and “HFD-non” abbreviations. The meaning of those abbreviations is not presented in the legend of the Figure. I suppose the correct abbreviations should be “STD-fed” and “HFD-fed”. 

Thank you for your comments and suggestions.

We have changed STD-non and HFD-non abbreviations to STD-Fed and HFD-Fed in all related legends (Fig 1, Fig 2, Fig 7, Fig 8, S3 Fig, S4 Fig, S5 Fig).We have also defined the abbreviations in the respective legends. 

Moreover, is there an explanation for the large variance noticed in the “HFD-non” group, which is not perceived in the other groups?

Thank you for your comments and suggestions.

To address this comment, we have revised the Result section.

We have added ‘albeit showing heterogeneity’ in the Results (P22 Line 374). We have also explained that ‘The HFD-fed mice at 20 weeks of age were in an early diabetic state and had elevated insulin secretion. The heterogeneity of insulin secretion by the HFD-fed mice at 20 weeks of age may be influenced by the ad libitum consumption of the HFD diet at night and differences in mouse-specific insulin resistance [26]. It is also possible that elevated postprandial blood glucose levels affect insulin secretion.’ (P23 Lines 378–382).

Lines 411 and 412: The authors demonstrated that phlorizin administration significantly increased serum triglyceride and total cholesterol levels in HFD-fed obese mice. Nevertheless, phlorizin supplementation has shown to reduce plasma cholesterol and tryglycerides in C57BL/6J HFD-fed mice in a previously published paper (Liu et al., 2021). I believe the authors should discuss these differences in the discussion.

Reference: Liu D, Ji Y, Guo Y, et al. Dietary Supplementation of Apple Phlorizin Attenuates the Redox State Related to Gut Microbiota Homeostasis in C57BL/6J Mice Fed with a High-Fat Diet. J Agric Food Chem. 2021;69(1):198-211. doi:10.1021/acs.jafc.0c06426

Thank you for your comments and suggestions.

To address this comment, we have revised the Discussion section as follows:

‘In HFD-fed obese mice, phlorizin administration significantly increased serum triglyceride and total cholesterol levels after a short administration. On the other hand, clearance of triglycerides and total cholesterol is known to be increased by SGLT inhibitors [35,36]. These results are consistent with the fact that in C57BL/6J HFD-fed mice, phlorizin supplementation decreases plasma cholesterol and triglycerides after long-term administration [37]’ (P33 Lines 566–571).

- Figure 8: To avoid confusion, I would suggest the authors to use similar colors for the same groups in Figure 8A and 8C. For instance, while blue is used for a HFD-PLZ group in Figure 8A, it is used for a HFD-vehicle group in Figure 8C. Using similar colors could avoid interpretation mistakes by the readers.

Thank you for your comments and suggestions.

To address this comment, we have revised Figure 8.

We have changed the colors following your advice to avoid misinterpretation by readers. Blue has been changed to yellow for the HFD-vehicle group, and red has been changed to blue for the HFD PLZ group in Fig 8C and 8D.

Discussion:

- Lines 827 to 829: The authors refer that one possible limitation of the study was that phlorizin is a non-selective SGLT inhibitor. Nonetheless, which SGLT do the authors believe is more related to the releasing of lamellar bodies from the apical membrane of PTECs?

Thank you for your comments.

To address this comment, we have revised the Discussion section as follows:

‘SGLT2 is predominantly expressed in the S2 segment, where lamellar bodies are abundant in HFD-fed mice, and it is possible that phlorizin mainly affected SGLT2 to induce the release of lamellar bodies from the apical membrane of PTECs.’ (P34-35 Lines 591–594)

Reviewer #2: Reviewers Comments

Manuscript ID: PONE-D-22-21058

Study Title: Treatment of tubular damage in high-fat-diet-fed obese mice using sodium-glucose co transporter inhibitors

Authors: Saitoh et al.

Authors: Saitoh et al.

The study entitled “Treatment of tubular damage in high-fat-diet-fed obese mice using sodium-glucose co-transporter inhibitors” by Saitoh et al aims to explore underlying nephroprotective effects mechanism of Sodium-glucose co-transporter inhibitors. The study is well designed and robust which shows the nephroprotective effects of lamellar bodies accumulated in the S2 segment and urinary excretion of sphingomyelin. The methodology is well described and results are well presented and discussed.

There are few minor concerns which needs to be taken care.

Minor Comments:

1. Page 11, line 173: Please reframe the sentence for clarity.

We are grateful for your useful comment. We have revised the Material Method section to address this. 

We have changed this sentence as follows: ‘Protocol 1: Phlorizin or vehicle (n = 34 in each STD and HFD group) was administered to mice at 2 and 16 h prior to perfusion fixation, organ resection, and blood collection.’ The designation of the time of administration of phlorizin was vague, so we have clarified it. 

2. The manuscript is overall too lengthy and needs to be concised wherever possible.

Thank you for your comment. We have made some changes to reduce the word count by submitting some of the figures and tables as supplementary materials. We have however retained the contents that are very essential for understanding our manuscript.

---

## [Editor Report · Decision Letter 1]

1 Feb 2023

Treatment of tubular damage in high-fat-diet-fed obese mice using sodium-glucose co-transporter inhibitors

PONE-D-22-21058R1

Dear Dr. Saitoh,

We’re pleased to inform you that your manuscript has been judged scientifically suitable for publication and will be formally accepted for publication once it meets all outstanding technical requirements.

Kind regards,

Ines Armando, PhD

Academic Editor

PLOS ONE
---

## [Editor Report · Acceptance letter]

3 Feb 2023

PONE-D-22-21058R1 

Treatment of tubular damage in high-fat-diet-fed obese mice using sodium-glucose co-transporter inhibitors 

Dear Dr. Saitoh:

I'm pleased to inform you that your manuscript has been deemed suitable for publication in PLOS ONE. Congratulations! Your manuscript is now with our production department. 

Kind regards, 

on behalf of

Dr Ines Armando 

Academic Editor

PLOS ONE